# Quantitative Analysis of the *MGMT* Methylation Status of Glioblastomas in Light of the 2021 WHO Classification

**DOI:** 10.3390/cancers14133149

**Published:** 2022-06-27

**Authors:** Levin Häni, Monika Kopcic, Mattia Branca, Alessa Schütz, Michael Murek, Nicole Söll, Erik Vassella, Andreas Raabe, Ekkehard Hewer, Philippe Schucht

**Affiliations:** 1Department of Neurosurgery, Inselspital, Bern University Hospital, University of Bern, 3010 Bern, Switzerland; monika.kopcic@spitalaarberg.ch (M.K.); alessa.schuetz@insel.ch (A.S.); michael.murek@insel.ch (M.M.); nicole.soell@insel.ch (N.S.); andreas.raabe@insel.ch (A.R.); philippe.schucht@insel.ch (P.S.); 2CTU Bern, University of Bern, 3012 Bern, Switzerland; mattia.branca@ctu.unibe.ch; 3Institute of Pathology, Inselspital, Bern University Hospital, University of Bern, 3010 Bern, Switzerland; erik.vassella@pathology.unibe.ch (E.V.); ekkehard.hewer@chuv.ch (E.H.); 4Institute of Pathology, Lausanne University Hospital and University of Lausanne, 1011 Lausanne, Switzerland

**Keywords:** glioblastoma, temozolomide, O(6)-methylguanine-DNA methyltransferase

## Abstract

**Simple Summary:**

The status of a DNA repair protein called *MGMT* is a prognostic marker in patients with glioblastomas, the most frequent malignant brain tumor. Epigenetic silencing of this gene predicts increased sensitivity to chemotherapy. Silencing is assessed by analyzing modifications, so-called methylation, of a certain gene region. Whereas most studies report such information only in a qualitative manner, quantitative testing might provide additional information. The aim of our study was to determine a quantitative threshold for better survival among patients with glioblastomas. Among 321 patients suffering from glioblastoma, we found better survival in patients with glioblastomas that have ≥16% methylation of a particular region of the *MGMT* gene. Above 16% methylation, we found no additional benefit with increasing degree of methylation. We suggest using this threshold for selection in clinical trials and for patient counseling.

**Abstract:**

Background: Glioblastomas with methylation of the promoter region of the O(6)-methylguanine-DNA methyltransferase (*MGMT*) gene exhibit increased sensitivity to alkylating chemotherapy. Quantitative assessment of the *MGMT* promoter methylation status might provide additional prognostic information. The aim of our study was to determine a quantitative methylation threshold for better survival among patients with glioblastomas. Methods: We included consecutive patients ≥18 years treated at our department between 11/2010 and 08/2018 for a glioblastoma, *IDH* wildtype, undergoing quantitative *MGMT* promoter methylation analysis. The primary endpoint was overall survival. Results: A total of 321 patients were included. Median overall survival was 12.6 months. Kaplan–Meier and adjusted Cox regression analysis showed better survival for the groups with 16–30%, 31–60%, and 61–100% methylation. In contrast, survival in the group with 1–15% methylation was similar to those with unmethylated promoter regions. A secondary analysis confirmed this threshold. Conclusions: Better survival is observed in patients with glioblastomas with ≥16% methylation of the *MGMT* promoter region than with <16% methylation. Survival with tumors with 1–15% methylation is similar to with unmethylated tumors. Above 16% methylation, we found no additional benefit with increasing methylation.

## 1. Introduction

Glioblastoma is the most common malignant primary tumor of the brain, with an incidence of 3.23/100,000 [1]. The standard of care consists of maximal safe resection followed by radiotherapy combined with temozolomide (TMZ) chemotherapy [2]. Despite considerable efforts to improve outcomes, the prognosis remains grim, with a median overall survival of 14–16 months [2,3,4,5].

Epigenetic silencing of the DNA repair protein O(6)-methylguanine-DNA methyltransferase (*MGMT*) has emerged as an important prognostic and predictive marker in patients with glioblastomas [6,7,8,9]. Methylation of specific cytosine–phosphate–guanine (CpG) sites within its promoter region silences the *MGMT* gene [10]. This, in turn, leads to increased sensitivity of the cancer cells to alkylating chemotherapy [8,9]. Therefore, the addition of TMZ chemotherapy improves survival in patients with tumors containing a methylated *MGMT* promoter [6]. However, for tumors with an unmethylated *MGMT* promoter, the benefit of TMZ is marginal and increasingly questioned [6]. A major obstacle in selecting patients for clinical trials according to their *MGMT* status is the lack of consensus regarding the method of determination and cut-off values for quantitative testing to classify *MGMT* promoters as unmethylated or methylated [11,12]. Whereas most studies report only qualitative information on the *MGMT* methylation status, quantitative testing might provide additional information and help select patients for further treatment. Most studies report a quantitative *MGMT* promoter methylation threshold between 1 and 30% to distinguish methylated from unmethylated status [11]. However, clarification of the optimal threshold of *MGMT* methylation below which TMZ can be omitted in experimental arms of clinical trials is needed [13]. In addition, knowledge of this threshold can help to stratify and balance patients for prognostic factors in future clinical trials.

Importantly, the diagnostic criteria for glioblastoma have shifted away from purely histomorphological toward molecular characteristics. In the 2021 WHO classification of tumors of the central nervous system, glioblastoma is defined by the absence of a mutation in the *isocitrate dehydrogenase* gene (i.e., *IDH* wildtype) [14]. So far, studies investigating the value of *MGMT* status have not applied this new definition.

The aim of our study was to determine a threshold of quantitative *MGMT* methylation status for better survival among patients with *IDH* wildtype glioblastomas. Through a subgroup analysis of tumors exposed to TMZ and comparison with a historical cohort, we sought a predictive threshold for benefit from TMZ. We hypothesized that even patients with a low percentage of *MGMT* promoter methylation benefit from chemotherapy.

## 2. Materials and Methods

### 2.1. Standard Protocol Approvals, Registrations, and Patient Consents

We conducted a retrospective, single-center observational study. Approval for this study was granted by the local ethics committee of the canton of Bern, Switzerland (2022-00698).

### 2.2. Patient Population

We screened consecutive patients treated for a glioma at our department between November 2010 and August 2018. Inclusion criteria were (i) diagnosis of glioblastoma, *IDH* wildtype according to the 2021 WHO classification of tumors of the central nervous system [14], (ii) availability of a quantitative *MGMT* promoter methylation analysis, and (iii) age > 18 years. The absence of an *IDH* mutation was determined by immunohistochemistry and/or sequencing. Importantly, gliomas were reclassified according to the 2021 WHO classification. Thus, we included diffuse and astrocytic tumors if the *IDH* status was wildtype and either a telomerase reverse transcriptase (TERT) promoter mutation, epidermal growth factor receptor (EGFR) gene amplification, or combined gain of chromosome 7 and loss of chromosome 10 was present. Exclusion criteria were (i) refusal of general consent for the scientific use of health-related data and (ii) missing information on *IDH* status or *MGMT* methylation status.

### 2.3. Data Analysis

The primary endpoint of the study was overall survival. For the analysis of the primary endpoint, we stratified patients into groups based on the results of the quantitative *MGMT* methylation analysis. Patients with 0% methylation constituting the reference population were compared with patients with 1–15%, 16–30%, 31–60%, and 61–100% *MGMT* promoter methylation.

To identify an *MGMT* methylation threshold for benefit from TMZ, we compared tumors exposed to TMZ in our cohort with a historical cohort treated by radiotherapy only. We used the upper bound of the confidence interval for median survival from the radiotherapy-only arm of the seminal trial by Stupp et al. to stratify patients into long-term (>13.0 months) or short-term (≤13.0 months) survivors [2]. Long-term survival was the secondary endpoint for the subgroup of patients with tumors exposed to TMZ.

Clinical data were extracted from our electronic Patient Data Management System. These included patients’ sex, age, preoperative Karnofsky performance status (KPS), date and type of surgery, pathological diagnosis, molecular pathology findings, and postoperative chemotherapy and/or radiotherapy. Postoperative MRIs were reviewed and classified as either complete resection of enhancing tumor (CRET), gross total resection with a minimal residual of <0.175 mL (GTR), subtotal resection with a residual of >0.175 mL (STR), or biopsy only.

We usually extracted DNA for *MGMT* promoter methylation analysis from the tumor center with >70% tumor cell density. Quantitative *MGMT* promoter methylation analysis was performed using a primer extension-based method adapted to formalin-fixed paraffin-embedded tissues, as previously described [15,16]. Six diagnostically relevant CpG dinucleotides, including CpG4, are mapped with this assay [8,9]. A multiplex primer extension was performed using fluorescent-labeled primers, which bind to CpG dinucleotides specific for the methylated or unmethylated form of the DNA, followed by capillary electrophoresis [15].

### 2.4. Statistics

Statistical analysis was performed using the statistical software Stata 17.0 (StataCorp. 2021. Stata Statistical Software: Release 17. College Station, TX, USA: StataCorp LLC). Descriptive statistics for continuous variables included calculation of the mean and standard deviation (SD). Comparison between groups was made using ANOVA. Categorical variables are described with frequencies and percentages. Between-group comparisons were made using a chi-square test.

The primary endpoint was analyzed using Kaplan-Meier curves of the different groups and comparing them using the log-rank test. The analysis was performed first in a univariable Cox regression (proportional hazards) model and then adjusted for baseline variables (age, sex, type of resection, KPS, radiotherapy, and chemotherapy) in a multivariable Cox model.

To identify the optimal cut-off value for a benefit from TMZ, we stratified patients according to whether they were long-term (>13.0 months) or short-term (≤13.0 months, corresponding to the historical radiotherapy-only group [2]) survivors. The percentage of methylation of each tumor was plotted on a receiver operating characteristics (ROC) curve with true positives (sensitivity) on the vertical axis and false positives (1—specificity) on the horizontal axis. The area under the curve (AUC) and its 95% confidence interval were calculated. The optimal cut-off point to differentiate the two survival groups was determined by calculating the Youden index, and the positive likelihood ratios for different levels of methylation were also compared.

We addressed missing values first by reanalyzing the source data or, if no value was retrievable, by pairwise deletion. Complete cases were considered for the Cox regression analysis. For all analyses, a *p*-value ≤ 0.05 was considered significant.

### 2.5. Data Availability

The study data are available and will be shared upon reasonable request from other investigators for the purposes of replicating results.

## 3. Results

### 3.1. Patient Population

We screened 885 patients in our local tumor database treated at our department between November 2010 and August 2018. Our final study population consisted of 321 patients with glioblastoma, *IDH* wildtype (Figure 1).

Of these 321 patients, 109 (34.0%) were female. Mean age at diagnosis (±SD) was 64.1 years (±11.4), and 108 patients (33.6%) were 70 years or older. A KPS < 70% was recorded in 56 patients (17.5%). A total of 224 (69.8%) patients underwent resection, and 97 (30.2%) patients were only biopsied. Of patients undergoing resection, postoperative MRI confirmed CRET in 123 (54.9%), GTR in 18 (8.0%), and STR in 78 (34.8%). Five patients (2.2%) did not undergo contrast-enhanced MRI within 48 h postoperatively. Surgery was followed by the combination of radio- and chemotherapy in 250 patients (77.9%), radiotherapy only in 28 patients (8.7%), and chemotherapy only two patients (0.6%). No further therapy was administered in 30 patients (9.4%). No information concerning radio- and chemotherapy was available for 11 patients (3.4%).

### 3.2. Impact of Quantitative MGMT Status on Survival

Patients were grouped according to their quantitative *MGMT* methylation status (Table 1). Between-group comparison yielded a significant difference for sex distribution (*p* = 0.003), with more female patients (71%) in the 1–15% methylated group, while all other groups contained more male patients. Other baseline variables did not show significant differences between groups.

Median overall survival was 12.6 months. We found an unmethylated *MGMT* promoter region in 165 patients (51.4%). Among these patients, median survival was 12.0 months. Quantitative *MGMT* status was associated with survival. Kaplan-Meier analysis showed a better survival for all groups with ≥16% methylation of the *MGMT* promoter region (Figure 2). Compared with patients with unmethylated tumors, Cox regression analysis showed significantly longer survival in the groups with 16–30%, 31–60%, and 61–100% methylation of the *MGMT* promoter (Figure 2, Table 2). In contrast, the group with 1–15% methylation displayed similar survival to that of patients with unmethylated promoter regions.

Other factors independently associated with overall survival were KPS ≥ 70% (HR 0.57, 95% CI 0.42–0.77, *p* < 0.001), CRET versus biopsy (HR 0.47, 95% CI 0.35–0.63, *p* < 0.001), chemotherapy (HR 0.38, 95% CI 0.25–0.59, *p* < 0.001), and radiotherapy (HR 0.31, 95% CI 0.18–0.51, *p* < 0.001). For GTR versus biopsy (HR 0.59, 95% CI 0.34–1.01, *p* = 0.053) and STR versus biopsy (HR 0.77, 95% CI 0.56–1.06, *p* = 0.114), there was a nonsignificant trend toward better survival. In contrast, sex (HR 0.90, 95% CI 0.70–1.17, *p* = 0.428) and age (HR 1.01, 95% CI 1.00–1.02, *p* = 0.261) showed no association with overall survival.

When analyzing only those patients not treated with TMZ (*n* = 69), no difference in survival was found among the different groups of *MGMT* methylation compared with patients with unmethylated tumors. However, the number of patients was low, and this result has to be interpreted cautiously.

In order to identify potential subgroups with better survival in the group with 1–15% methylation, we performed a post hoc sensitivity analysis comparing patients with unmethylated tumors to tumors with 1–7% methylation, 8–15% methylation, and ≥16% methylation. Survival of patients with tumors with 1–7% methylation (HR 1.10, 95% CI 0.49–2.49, *p* = 0.821) and 8–15% methylation (HR 1.59, 95% CI 0.74–3.39, *p* = 0.233) was similar to unmethylated tumors, whereas patients with tumors with ≥16% methylation displayed significantly better survival (HR 0.50, 95% CI 0.39–0.64, *p* < 0.001) (Figure 3).

### 3.3. Predictive Cut-Off of Quantitative MGMT Status

TMZ was administered to 252 patients. Of these, 149 (59.1%) reached our secondary endpoint of long-term survival compared with the historical radiotherapy-only cohort (>13.0 months). Although only 52.4% of patients with an unmethylated *MGMT* promoter region and 42.9% with 1–15% methylation reached long-term survival, 78.6%, 63.2%, and 68.7% in the 16–30%, 31–60% and 61–100% methylation group, respectively, did so (*p* = 0.023).

Plotting the percentage of methylation for each patient on a ROC curve using long-term survival as the outcome parameter yielded an AUC of 0.59 (0.52–0.65). The Youden index was maximal at ≥16% methylation of the *MGMT* promoter with an associated positive and negative likelihood ratio of 1.536 and 0.712, respectively. The sensitivity and specificity for long-term survival with a cut-off of ≥16% were 54% and 65%, respectively.

## 4. Discussion

Our results show a signal toward better survival of patients with tumors that have ≥16% methylation of the *MGMT* promoter region. In a secondary analysis, 16% was found as the best-estimated cut-off for increasing sensitivity to TMZ.

### 4.1. Prognostic Threshold of Quantitative MGMT Testing

The strong prognostic value of the qualitative assessment of the methylation status of the *MGMT* promoter region has been repeatedly demonstrated [2,5,6,7,17]. A meta-analysis of 34 studies found a significantly better overall survival of patients with methylation of the *MGMT* gene promoter, with a pooled hazard ratio of 0.494 [7].

In contrast, studies investigating a quantitative threshold are sparse. A cut-off of >8–9% methylation of five CpG sites detected by pyrosequencing has been associated with better survival [18,19,20]. Dunn et al. reported that the best threshold for better overall survival is >29% methylation [21]. However, even tumors with >9 to ≤29% methylation showed a survival benefit in their analysis. Reifenberger et al. reported that the survival of patients with 8–25% methylated tumors was worse and similar to those with <8% methylated tumors, which led them to propose a cut-off of 25% methylation [22]. By using pyrosequencing, Nguyen et al. found an optimal cut-off of 21% methylation for better survival [23]. In line with our results, Yuan et al. reported an optimal cut-off of 12.5% to stratify patients as good and poor responders to TMZ [24]. Considering our results together with the existing literature, we suggest a cut-off for better survival in quantitative *MGMT* methylation analysis between a very low and low degree of methylation of around 16%. In contrast, we found no survival advantage among patients with 8–15% methylation compared with 1–7% methylation or unmethylated tumors.

### 4.2. MGMT Promoter Methylation: The More, the Better?

Dunn et al. reported that patients with tumors with 10–20% methylation survived longer than those with unmethylated tumors, and the effect was even more pronounced with a higher degree of methylation [21]. Specifically, survival was better if tumors were >29% methylated than when tumors were >9 to ≤29% methylated. However, these results should be interpreted with caution because the >29% methylated group consisted of only 25 patients with a median survival of 26.2 months, indicating the potential for selection bias in this group. In contrast, we found a similar survival of patients with tumors with 16–30%, 31–60%, and 61–100% methylation. Likewise, in a pooled analysis of four large trials, Hegi et al. found that more methylation is not associated with better outcomes above a certain degree of methylation [13]. Compared with patients with unmethylated tumors (corrected *MGMT* log_2_ ratio ≤ −0.28), they reported a significant survival benefit for patients in a gray zone (corrected *MGMT* log_2_ ratio > −0.28 to ≤1.27) and for patients with a corrected *MGMT* log_2_ ratio > 1.27 [13]. In accordance with our findings, Reifenberger et al. reported no difference in survival between patients with 26–50% and >50% methylated tumors [22]. Hence, we suggest that above a certain threshold, more *MGMT* promoter methylation is not associated with better survival.

### 4.3. Can TMZ Be Omitted According to MGMT Promoter Methylation?

The efficacy of TMZ for patients with *MGMT* unmethylated tumors is, at best, limited. In a pivotal trial in 2005, when treated with TMZ and radiotherapy, these patients had only a marginal, nonsignificant advantage when compared with radiotherapy only [2,6]. Importantly, later analysis demonstrated an *IDH*1 mutation in 7% of patients, signifying a prognostic factor not considered in the initial study [25]. The finding that low-grade gliomas with an *IDH* mutation frequently have a methylated *MGMT* promoter could be extrapolated to high-grade gliomas, which might potentially explain the nonsignificant trend in this study [26,27]. Consequently, we defined the confirmation of *IDH* wildtype as an inclusion criterion for our study.

Evidence supporting the futility of TMZ for patients with an unmethylated *MGMT* promoter is provided in the GLARIUS trial, in which no difference in overall survival was found between patients with unmethylated tumors treated with TMZ and patients in the experimental arm treated with an *MGMT* promoter methylation independent chemotherapy (irinotecan) [3]. Hence, the omission of alkylating chemotherapy in patients with unmethylated tumors is not detrimental.

A cohort study of 233 glioblastoma patients aged ≥70 years found no survival benefit of TMZ combined with radiotherapy over radiotherapy alone among those with unmethylated tumors [22]. In contrast, patients with methylated tumors had longer progression-free survival when treated with radiotherapy plus TMZ or TMZ alone compared with patients receiving radiotherapy alone.

Interestingly, the recently published CATNON trial found no survival benefit of either concurrent or adjuvant TMZ among patients with *IDH*1 and *IDH*2 wildtype high-grade gliomas [28]. Although the trial included anaplastic gliomas rather than glioblastomas, many of the *IDH* wildtype anaplastic gliomas would now be reclassified as glioblastomas according to the 2021 WHO classification. However, the limited number of patients with *IDH* wildtype tumors in this study means that no definitive conclusions can be drawn. A further report of the association of the *MGMT* status with outcome is pending.

To sum up, despite limited efficacy, the omission of TMZ in adult patients <70 years with a good KPS and an *MGMT* unmethylated tumor should be restricted to clinical trials. As suggested by Hegi et al., patients whose tumors have a low degree of *MGMT* methylation (gray zone) may still gain some benefit from TMZ [13]. Consequently, the lower safety margin is suitable for the inclusion of patients with truly unmethylated tumors in clinical trials omitting TMZ. We propose a threshold of <16% of *MGMT* methylation for inclusion in such a trial.

### 4.4. Method for MGMT Methylation Assessment

There is no consensus concerning the optimal laboratory method to determine *MGMT* methylation status for clinical decision-making [29]. Methylation-specific polymerase chain reaction (mPCR) and pyrosequencing are most commonly used and provide the most accurate prognosticators [11,12,18]. However, there is considerable disagreement about which CpG sites in the *MGMT* promoter region should be analyzed [11,12]. The number of CpG sites analyzed varies from 1–3 up to more than 16 and is often determined by the testing kit manufacturer [11]. Particularly in cases with the heterogenous methylation of individual CpG sites, the degree of methylation might depend much on the method used. However, different regions of methylation throughout the *MGMT* gene have different prognostic values [30]. Unfortunately, it remains uncertain which CpG islands have the highest clinical value [12,29]. A particular CpG site in exon 1 (termed CpG4) has been suggested as particularly important for the correlation with survival [18]. Our primer extension-based method for quantitative assessment of the methylation status of the *MGMT* promoter maps six diagnostically relevant CpG dinucleotides, including the aforementioned CpG4 site [6,8,9,15,18,31]. The method used has been shown to be sensitive and consistently reproducible using routinely processed tissue samples [15]. Compared with quantitative mPCR, our primer extension-based PCR may better estimate the intratumoral heterogeneity of *MGMT* methylation [15]. In addition, quantitative mPCR depends on two methylation-specific primers for amplification and, thus, requires more sites to be methylated compared with primer extension-based PCR [15].

Arguably, pyrosequencing might be a better method for *MGMT* promoter methylation analysis due to its high reproducibility, high sensitivity, and high resolution for individual CpG sites [12]. However, our method yielded a similar cut-off to the one reported by Yuan et al. from using pyrosequencing [24]. Moreover, Reifenberger et al. reported a strong concordance between pyrosequencing and mPCR using a cut-off of 8% for pyrosequencing [22].

### 4.5. Clinical Implications

Our results have clinical implications beyond their benefit for clinical trials. In the NOA-08 trial, elderly patients with high-grade gliomas with *MGMT* promoter methylation displayed a longer progression-free survival when treated with TMZ alone compared with radiotherapy alone (8.4 versus 4.6 months), whereas patients with unmethylated tumors had a longer progression-free survival with radiotherapy alone compared with TMZ alone (4.6 versus 3.3 months) [32]. Similarly, the Nordic trial found a significantly longer overall survival of patients with *MGMT* methylated compared with unmethylated tumors when treated with TMZ alone but no difference in overall survival when treated with radiotherapy alone [33]. The median overall survival of patients with *MGMT* promoter methylation in this trial was 9.7 months (95% CI 8.0–11.4) when treated with TMZ alone compared with 8.2 months (95% CI 6.6–9.9) when treated with radiotherapy alone [33]. Hence, these trials consistently suggest that elderly patients eligible for either radiotherapy or TMZ should undergo *MGMT* testing before clinical decision-making [34]. TMZ has an acceptable safety profile in elderly patients with poor performance status, and TMZ treatment leads to a significant improvement in functional status and an increased survival compared with supportive care alone [35]. Our data suggest that in patients with tumors with <16% methylation of the *MGMT* promoter region, TMZ may be forgone or discontinued, particularly when poorly tolerated, for instance, in elderly patients.

Likewise, *MGMT* status impacts the management of patients with recurrent glioblastomas. For *MGMT* methylated tumors, rechallenging with TMZ or switching to another alkylating agent such as CCNU can be a good option [36]. We propose to use ≥16% methylation of the *MGMT* promoter region as a threshold for rechallenging with TMZ, while alternative strategies should be sought for those with <16% methylation.

In addition, we found no difference in survival between the methylation groups among patients unexposed to TMZ. This result is relevant for improved patient counseling at diagnosis and suggests that the benefit in the survival of methylated tumors is largely transferred through TMZ. However, due to the limited number of patients, the significance of this result is questionable and should be interpreted cautiously.

### 4.6. Limitations

Several limitations apply to our study. Firstly, our results were obtained by a retrospective analysis of data from a single center. Secondly, the method used for *MGMT* analysis in our center has not gained widespread use. Nevertheless, our method has been shown to be sensitive and consistently reproducible [15]. Thirdly, the degree of methylation also depends on the tumor cell content of the sample analyzed. We tried to extract DNA from the tumor center with >70% tumor cells. However, suboptimal sampling with fewer tumor cells might lead to an erroneous interpretation of the results. Fourthly, even though our secondary analysis confirmed the results of the primary analysis with a threshold of ≥16%, the predictive power of the model is low, with an AUC of 0.59 (0.52–0.65). A potential explanation is the modest benefit of TMZ for overall survival in glioblastoma patients. TMZ translates into a median overall survival benefit of only 2.5 months [2]. Hence, the survival curves of responders and nonresponders to TMZ largely overlap. This huge overlap explains the poor discriminatory power of the *MGMT* analysis for long-term survival (≥13.0 months) and, in turn, the low AUC, sensitivity, and specificity. Furthermore, the Youden index used to determine a threshold in our secondary analysis puts equal weights on sensitivity and specificity. However, in a clinical scenario, we would recommend a more conservative approach, i.e., to put more weight on sensitivity than specificity. In other words, to administer TMZ despite futility would be a lesser evil than to withhold TMZ despite activity.

## 5. Conclusions

Patients with glioblastomas with ≥16% methylation of the *MGMT* promoter region display improved survival compared with those with tumors with <16% methylation. The survival of patients with tumors with 1–15% methylation is similar to that of those with unmethylated tumors. Above 16% methylation, we found no additional benefit with increasing degree of methylation. We suggest using the 16% threshold for *MGMT* methylation for selecting patients for TMZ treatment in clinical trials and for prognostic counseling.

## Figures and Tables

**Figure 1 cancers-14-03149-f001:**
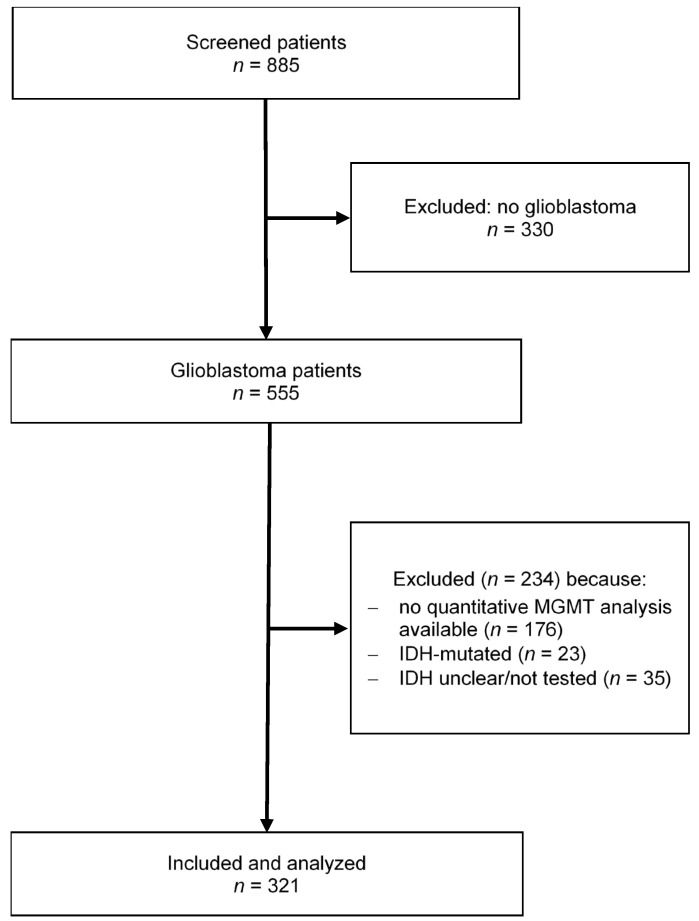
Flowchart illustrating patient selection.

**Figure 2 cancers-14-03149-f002:**
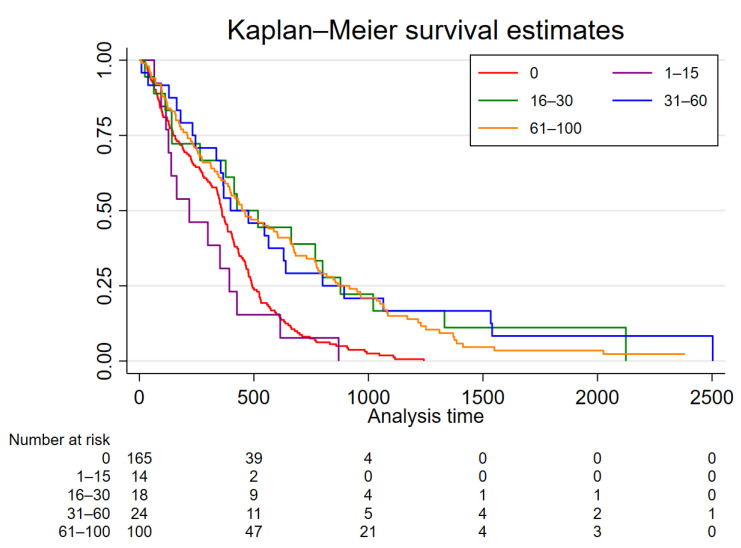
Kaplan-Meier analysis stratified for *MGMT* promoter methylation status. Patients with ≥16% methylation of the *MGMT* promoter region displayed significantly longer survival compared with patients with unmethylated tumors. Survival of patients with 1–15% methylation was similar to that of patients with unmethylated *MGMT* promotors.

**Figure 3 cancers-14-03149-f003:**
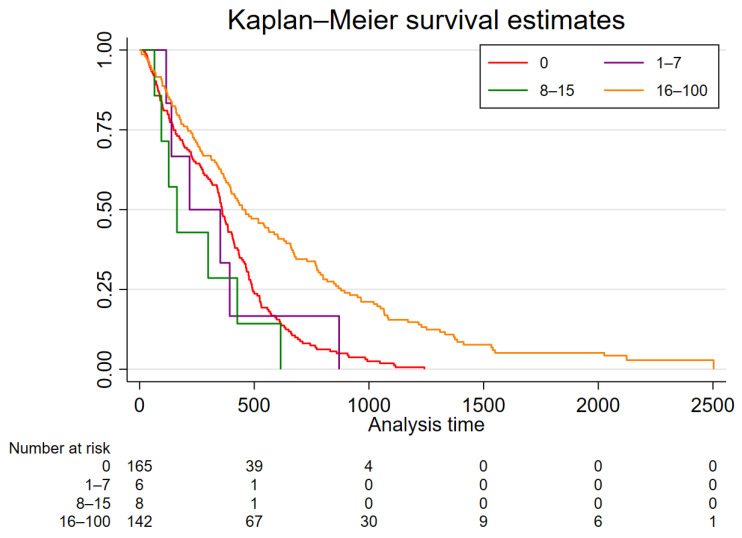
Kaplan-Meier subgroup analysis of patients with low *MGMT* promoter methylation. Survival of patients with tumors with 1–7% and 8–15% methylation of the *MGMT* promoter region showed no difference and was similar to patients with unmethylated tumors. Compared with these subgroups, survival of patients with tumors with ≥16% methylation was longer.

**Table 1 cancers-14-03149-t001:** Comparison of baseline variables by *MGMT* methylation group.

Percentage Methylation	0% *n* = 165	1–15% *n* = 14	16–30% *n* = 18	31–60% *n* = 24	61–100% *n* = 100	*p*-Value
**Female sex**	43 (26%)	10 (71%)	6 (33%)	8 (33%)	42 (42%)	0.003
**Age (years)**	63 (±12)	61 (±9.3)	67 (±8.7)	62 (±13)	66 (±10)	0.06
**KPS ≥ 70%**	139 (84%)	10 (71%)	18 (100%)	18 (75%)	80 (80%)	0.15
**Surgery**						0.44
**-biopsy**	48 (29%)	5 (36%)	7 (39%)	8 (33%)	29 (29%)	
**-CRET**	70 (42%)	4 (29%)	7 (39%)	4 (17%)	38 (38%)	
**-GTR**	11 (6.7%)	0 (0%)	0 (0%)	2 (8.3%)	5 (5%)	
**-subtotal**	33 (20%)	5 (36%)	3 (17%)	9 (38%)	28 (28%)	
**Chemotherapy**	126 (76%)	10 (71%)	14 (78%)	19 (79%)	83 (83%)	0.72
**Radiotherapy**	141 (85%)	12 (86%)	14 (78%)	21 (88%)	90 (90%)	0.66

Between-group comparisons yielded a significant difference in the sex distribution among the different *MGMT* methylation groups. Absolute numbers and percentages are given for nominal variables. Means and standard deviations are given for continuous variables: CRET, complete resection of enhancing tumor; GTR, gross total resection with a minimal residual (<0.175 mL); KPS, Karnofsky performance status.

**Table 2 cancers-14-03149-t002:** Survival according to quantitative *MGMT* promoter methylation status.

Promoter Methylation	Unadjusted HR (95% CI)	*p*-Value	Adjusted HR (95% CI)	*p*-Value
**0%**	reference	n/a	reference	n/a
**1–15%**	1.32 (0.75–2.32)	0.340	1.47 (0.81–2.66)	0.204
**16–30%**	0.47 (0.28–0.78)	0.004	0.50 (0.29–0.85)	0.010
**31–60%**	0.45 (0.28–0.72)	0.001	0.39 (0.25–0.63)	<0.001
**61–100%**	0.52 (0.40–0.67)	<0.001	0.54 (0.41–0.72)	<0.001

On univariable analysis, survival was significantly better for patients with >15% methylation of the *MGMT* promoter region. In contrast, no difference between patients with unmethylated and 1–15% methylated tumors was found. In a multivariable analysis, HRs were adjusted for age, sex, KPS, resection status, and chemo- and radiotherapy: 95% CI, 95% confidence interval; HR, hazard ratio; KPS, Karnofsky performance status.

## Data Availability

The study data are available and will be shared upon reasonable request from other investigators for the purposes of replicating results.

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
