# Peer review of "Quantitative Analysis of the MGMT Methylation Status of Glioblastomas in Light of the 2021 WHO Classification"

_cancers, 2022, doi:10.3390/cancers14133149_

Round 1

Reviewer 1 Report

This study aimed to determine a quantitative MGMT promotor methylation threshold for better survival among patients with isocitrate dehydrogenase (IDH) wildtype glioblastomas. The authors found better survival of patients with glioblastomas that have ≥ 16% methylation of a particular region of the MGMT gene and suggest using this threshold for selection in clinical trials and for patient counselling.

In the current context of the study topic, the article is well chosen and the scientific relevance is high considering the current world situation regarding the treatment and management of glioblastomas. The topic of the article is original and relevant, the research aims of this manuscript are good and provide an advance in current knowledge in the field.

The article complements the existing published literature and offers a different threshold for MGMT promotor methylation status, discussing it in relation to the existing literature. I would recommend the authors to consult also https://doi.org/10.2217/cns-2021-0002, maybe it was omitted in the literature search. This topic is subject to variation between research groups and patient cohorts, so it is important to have additional research on the subject. There is also a paragraph stating the limitations and strengths of this study, which is most welcome.

The study is rationally designed and the methods are described in sufficient detail. I don’t have any specific suggestions to improve the methodology.

The manuscript is well written and clear, and the information is well organized and discussed. The discussion and conclusions are very well written, coherent and supported by the results. The conclusions address the main question of the study.

The references are appropriate and current.

The tables are clear and easy to read. There is also a paragraph stating the limitations and strengths of this study, which is most welcome. Minor spell check required. The Kaplan-Meier figures should be improved regarding the representation (the individual lines are difficult to distinguish since they are mostly overlapped). Perhaps lines of different colours would be better?  

Author Response

Dear reviewer

We thank you for your work and your suggestions. 

As proposed, we added a color-coding of lines in figure 2 and 3 to improve visibility. Moreover, we added the the work of Nguyen et al. to the discussion and the reference list.

Kind regards

L. Häni

Reviewer 2 Report

Methylation of MGMT promoter in glioblastoma is a prognostic marker used to predict increased sensitivity to treatment however, the decision about radiotherapy plus TMZ treatment does not take into account a quantitative methylation threshold for better survival among patients.
Authors are conscious of the limitations of their work; however, it represents the first step to proving it in other clinical cohorts to establish a threshold of MGMT methylation that could help in patients treatment.
Many works have been done in the last 15 years establishing MGMT promoter methylation as a prognostic marker. However, this work results original to define which % is the mínimum to consider MGMT promoter methylation as a prognosis treatment marker.
 Conclusions are consistent with the evidence and arguments
 The references are appropriate to the work.
 In Kaplan Meier curves, I would appreciate it if you could change the symbols on it because it is hard to follow in groups 16–30%, 31–60%, and 61–100% MGMT promoter methylation.

Author Response

Dear reviewer

We thank you for your work and your suggestions.

As proposed, we added a color-coding of lines in figure 2 and 3 to improve visibility. 

Kind regards

L. Häni